# The Use of Bi-Nasal Prongs for Delivery of Non-Invasive Ventilation to Foals

**DOI:** 10.3390/ani14060865

**Published:** 2024-03-11

**Authors:** Michael van Diggelen, Chris T. Quinn, Chee Sum M. Catanchin, Heidi S. Lehmann, Sharanne L. Raidal

**Affiliations:** Veterinary Clinical Centre, Charles Sturt University, Wagga Wagga, NSW 2678, Australia; michael.vandiggelen@gmail.com (M.v.D.); cquinn@csu.edu.au (C.T.Q.); ccatanchin@csu.edu.au (C.S.M.C.); hlehmann@csu.edu.au (H.S.L.)

**Keywords:** bi-level positive airway pressure (bi-PAP), continuous positive airway pressure (CPAP), equine critical care, equine respiratory physiology, neonatology, non-invasive ventilation (NIV), pressure support ventilation (PSV), respiratory insufficiency, respiratory support

## Abstract

**Simple Summary:**

Non-invasive ventilation (NIV) is a method of providing respiratory support without the need for airway intubation. In human patients, NIV can be delivered through a helmet, a mask or nasal prongs. The current study was undertaken to assess whether foals tolerate nasal prongs for the delivery of NIV. Bi-nasal prongs were well tolerated by healthy, standing foals, remaining in place for the allocated five minutes in four of six unsedated foals and, subsequently, in five of six lightly sedated foals. All foals tolerated NIV through bi-nasal prongs, although increasing airway pressures were associated with increased air leakage and discomfort in most foals, prompting discontinuation of treatment. These results suggest that bi-nasal prongs might be suitable for NIV in foals but that design or fitting requires further optimization and, in clinical cases, the lowest possible pressures should be used commensurate with necessary respiratory support.

**Abstract:**

Non-invasive ventilation (NIV) is a method of providing respiratory support without the need for airway intubation. The current study was undertaken to assess tolerance to bi-nasal prongs and NIV in healthy, standing, lightly sedated foals. Bi-nasal prongs were well tolerated by foals, remaining in place for the allocated five minutes in four of six unsedated foals and, subsequently, in five of six lightly sedated foals. All foals tolerated NIV through bi-nasal prongs, although increasing airway pressures were associated with increases in inspiratory volume, duration of inspiration and air leakage in most foals. These changes preceded discontinuation/intolerance of NIV on the basis of behaviour changes consistent with discomfort. Increased circuit leakage was associated with reduced return of expired air to the ventilator and increasing disparity between inspiratory and expiratory times and tidal volumes. The study results suggest that bi-nasal prongs might be suitable for NIV but that design or fitting requires further optimization and that behaviour and ventilator variables should be monitored to assess patient tolerance of the procedure.

## 1. Introduction

Respiratory insufficiency in foals occurs due to developmental, infectious and inflammatory conditions and can be due to pulmonary and non-pulmonary dysfunction. In foals, clinical manifestations of respiratory insufficiency include elevated respiratory rate, exaggerated intercostal and/or abdominal effort, abducted elbows, flared nostrils, extended head and neck position [1]. In a dyspneic foal, the work of breathing (WOB) is increased as respiratory muscles fatigue [2], and WOB may be increased in recumbent neonates. The consequences of these conditions include hypoxemia (PaO_2_ < 45 mmHg), hypercapnia (PaCO_2_ > 50 mmHg) and acidemia [3]. Inadequate gas exchange can occur due to inadequate ventilation, diffusion limitations, ventilation–perfusion mismatching or vascular shunting, resulting in type 1 (hypoxemia with normal or reduced PaCO_2_) or type 2 (hypercapnia) respiratory failure.

In foals, respiratory insufficiency is commonly addressed by nasal oxygen insufflation, which will satisfactorily address type 1 respiratory failure but does not improve ventilation. Mechanical ventilation is invasive, expensive and not readily available in most practice settings. Respiratory support of human neonates is now largely dependent on non-invasive ventilation (NIV) techniques [4,5], and preliminary studies have suggested that this modality is a promising option for the management of respiratory insufficiency in foals [6,7,8]. 

In human neonates, the optimal delivery and effectiveness of NIV respiratory support measures depend primarily on the patient-device interface (PDI) and the pressure-generating source [9]. There is considerable contention within the medical literature regarding the best PDI for NIV [10]. The ideal PDI ensures accurate and consistent pressure delivery, creates an effective seal to minimize air leak and minimizes the risk of iatrogenic nasal injury. Common NIV PDIs utilized in the delivery room in human neonates include round and anatomical masks; single, bi-nasal or nasopharyngeal prongs; nasal mask; and nasal cannulas [5]. Current consensus guidelines for the management of respiratory distress syndrome recommend short bi-nasal prongs or a mask interface as a first-line respiratory support method in human neonates [11]. To date, studies in foals have used mask delivery [6,7] for NIV. However, mask administration of NIV has necessitated constant and direct supervision of the foal (an attendant holding the mask in place to ensure that leakage is minimized and that nostrils are not occluded), and researchers have considered possible adverse consequences of increased equipment dead space and potential CO_2_ accumulation within the mask [7]. Nasotracheal intubation has been used in an interventional study on healthy foals with pharmacologically induced respiratory insufficiency to characterize respiratory effects associated with positive pressure ventilation [8], but the avoidance of intubation is one of the greatest benefits of NIV in clinical settings. More recently, the use of nasopharyngeal canulae has been described for the delivery of high flow oxygen therapy in foals [12], but this device was considered unlikely to provide an adequate seal for NIV. 

The current study was designed to assess tolerance of a prototype bi-nasal prong PDI by healthy foals with and without light sedation; to assess the tolerance of NIV delivered by nasal prongs in standing, lightly sedated foals; and to use spirometry to compare mask and nasal prong PDI’s in sedated and non-sedated foals. Our hypotheses were that nasal prongs and NIV would be poorly tolerated by foals and that spirometry would not demonstrate differences between nasal prongs and mask PDIs. 

## 2. Materials and Methods

### 2.1. Animals

Six Connemara-cross foals (out of Thoroughbred mares) from the Charles Sturt University teaching herd were available for inclusion. The three colts and three fillies ranged in age from 3 to 5 days (mean 4 days) and weighed 45 to 58 kg (mean 52.8 kg). All foals were born observed but unassisted in the paddock at term, and no foal had any recognized gestational or periparturient disorders. All foals had serum IgG > 8 g/L within 24 h of birth, and all were normal on veterinary examination prior to recruitment into the study. The study protocol was approved by the Charles Sturt University Animal Care and Ethics Committee (ACEC number 20208).

### 2.2. Experimental Design, and Physiological and Behavioural Responses

This study was conducted as a prospective interventional study with interventions conducted in a sequential manner (Table 1). Mares and foals were placed in an indoor loose box, with a separate but adjacent enclosed area where the foal could be treated in close proximity to the mare (Appendix A). After an initial physical examination, the foal was moved into the dedicated space next to the mare. If required, mares were lightly sedated (xylazine 0.4 mg/kg and acetylpromazine 0.02 mg/kg intravenously); other mares stood quietly or were offered a small feed adjacent to the foal. 

With the foal restrained, telemetric electrocardiography electrodes and transmitters (ECG; Televet Telemetric ECG and Holter, Engel Engineering Service GmbH, Heusenstamm, Germany) were placed on the foal’s chest and secured with gauze and adhesive bandage, as described previously [6]. Heart rate (HR) was taken from telemetry ECG displays at the end of each observation window. Foal respiratory rate (RRobs) was documented by observation (chest and/or nostril movement), and respiratory effort was subjectively assessed, during the final 30 s of each intervention. 

A novel ethogram was constructed to assess behavioural responses to sequential interventions. Briefly, researchers identified behaviours considered indicative of distress or discomfort associated with novel apparatus. Nominated behaviours were based on the combined clinical and horse-handling experience of the research team and limited available literature [13,14]. Using this ethogram, the number of times each foal exhibited a behaviour within consecutive 1 min observation windows was documented in real time based on direct observation by one researcher (MVD). As the exploratory nature of the current study precluded validation of the ethogram prior to use, only the frequency of each behaviour was recorded; no attempt was made to score the intensity of the observed behaviour/s nor to allocate a measure of distress (weighting) to each item. The utility of the document was reviewed by the research team immediately on completion of interventions for each foal, resulting in the final format (Figure 1), which varied little from the original document (Appendix A). 

Subsequently, video footage (Appendix A) of each foal was reviewed in random order by a second researcher (SLR) to independently complete the revised ethogram and assign a sedation score (Table 2), adapted from a previous publication [15]. For each item relating to sedation, specific responses were scored from 0 to 3 and the maximum values for each criterion observed during each intervention were summed to give a maximum score of 12. Although video sequences were analysed in random order, foal identity and intervention were, to some degree, self-evident. Where a discrepancy was apparent between direct and retrospective evaluation, contemporaneous notes and video footage were reviewed by the research team to determine the final ethogram result. In addition, based on informal observations during data acquisition, video footage was used to identify foals displaying periods of somnolence, defined as the foal standing quietly with absence of movement and with progressive lowering of the head, relaxed ear carriage and narrowed palpebral fissure, often to the extent of eye closure (Appendix A). In some cases, the foal was roused when it fell or nearly fell over. The number of somnolent episodes within each one-minute observation window was documented for all interventions. Interventions were discontinued if the foal showed intense or repeated behaviours indicating distress, or if there was any evidence of respiratory distress. 

Following initial observations (T1 and T2), foals were sedated with 5 mg of diazepam by intravenous injection. Subsequent observations (T3) commenced within 1 to 2 min of injection. Bi-nasal prongs were inserted with dummy tubing (Figure 2) in unsedated foals at T2 and following sedation at T4. 

### 2.3. Instrumentation

A prototype bi-nasal prong device (Figure 2) was fashioned using 10 mm cuffed endotracheal tubes cut down to approximately 8 cm length, with care to preserve the cuff tubing. Prongs were used with moderate (~60–80%) inflation of the cuffs, achieved by insufflation with 6 to 8 mL of air. Modified endotracheal tubes were connected directly to 13 mm propylene three-way irrigation connectors from a local hardware supplier. Prongs were joined centrally using 2 cm lengths of 10 mm vinyl tubing and connected to dummy tubing (12 cm inspiratory limb and 15 cm expiratory limb) at either end by 15 mm to 22 mm elbow connectors. A vented non-rebreathing valve (Fisher and Paykel Healthcare Limited, Auckland, New Zealand) was incorporated into the expiratory limb, which was always placed to the right of the foal’s face and secured using a modified harness (Figure 2). NIV was delivered at incremental pressure support (PS) and positive end-expiatory pressure (PEEP) by connection of ventilator tubing directly to elbow connectors on the bi-nasal prongs after removal of the non-rebreathing valve.

Non-invasive ventilation was delivered using a Mek MTV1000 transport ventilator (MEKICS Co., Paju City, Republic of Korea; distributed by Mediquip Pty Ltd., Loganholme, Australia) connected to the elbow adaptors (Figure 2), using default settings for spontaneous breathing mode (pressure-assisted ventilation in the event of apnoea, inspiratory flow triggering at 7 L/min) and room air. Pressure support (PS) and positive end-expiratory pressure (PEEP) were both set initially at 2 cmH_2_O (T5), and both settings were increased progressively by 2 cmH_2_O increments every two minutes for subsequent interventions (Table 1). Ventilator data including inspiratory and expiratory tidal volume (Vti and Vte, respectively), expiratory minute ventilation (VEmin), respiratory rate (RRvent), mean and peak airway pressure (Pmean and Ppeak), intrinsic PEEP (autoPEEP, the difference between measured and configured PEEP), inspiratory and expiratory time (Ti and Te) were obtained from video recordings of the ventilator display after completion of all interventions. The results were recorded as the mean value derived from five representative, but not necessarily consecutive, breaths during the final 30 s of each intervention.

### 2.4. Spirometry and Gas Analysis

Spirometry was performed at T1 (mask without sedation), T2 (nasal prongs without sedation), T3 (mask after sedation) and T4 (nasal prongs after sedation) after 5 min observations (T1 and T3) or after completion of 5 min intervention windows (T2 and T4). As previously described [7], a large veterinary anaesthesia mask (SurgiVet large canine mask, product number 32393B1; Sound Veterinary Equipment, Rowville, Australia) was manually held on the foal’s muzzle and connected to a respiratory flow head (Respiratory Flow Head 300 L, MLT300L, ADInstruments, Bella Vista, Australia) and gas sampling port (Figure 3) after 5 min observation at T1 and T3. The dead space of this apparatus was 60 mL (measured by water displacement). Data were collected for 30 to 60 s (sufficient to ensure 10 artifact-free breath cycles) immediately following completion of each 5 min window and analysed using PowerLab 4/25, Gas Analyser ML206 and LabChart 8 software (ADInstruments, Bella Vista, Australia), as previously described [6]. Tidal volume (Vt), respiratory frequency (Rf), peak inspiratory and peak expiratory air flow (PIF and PEF), and the duration of inspiratory (Ti) and expiratory (Te) phases were determined by post-sampling analysis of six consecutive and artifact-free breath cycles representative of tidal breathing. Spirometry, and inspired and expired gas analyses (FiO_2_, FeO_2_, FiCO_2_ and FeCO_2_) were performed following calibration of the spirometer pod using a seven-litre certified calibration syringe (Hans Rudolph Incorporated, Shawnee, KS, USA), and the gas analyser was calibrated using a two-point calibration of room air (20.9% O_2_, 0.04% CO_2_) and carbogen (95% O_2_, 5% CO_2_; BOC Gas, Wagga Wagga, Australia).

An adaptor was created to connect the flow head and gas sampling port to bi-nasal prongs after 5 min observation at T2 and T4. The adaptor consisted of anaesthesia circuit connectors, 13 and 19 mm irrigation connectors (total length 28 cm from the elbow connectors). The adaptor was attached to the flow head using a rubber spacer (Figure 3). Dead space of this apparatus was 75 mL. Spirometry data were collected and analysed as described for foals using masks by disconnecting the dummy tubing, rotating the elbow connectors 180° and connecting the adaptor to the bi-nasal prongs. The foal’s head was cradled using one arm and hand, and the flow head was supported in the opposite hand. Foals were allowed to recover for 1 to 2 min following the collection of spirometry data, prior to the collection of behavioural data during the next intervention. 

### 2.5. Statistical Methods 

A power analysis based on previous spirometry studies suggested that a sample size of six foals would discriminate differences in peak expiratory flow of 0.4 m/s, assuming standard deviation of 0.5 m/s, power of 80% and *p* = 0.05. The outcome variables included physiological responses (HR and RRobs), behavioural responses, spirometry values and ventilator data. Individual behavioural responses were combined into a composite ethogram and evaluated qualitatively. A single sedation score was recorded for each intervention window, and changes were evaluated using a Kruskal–Wallis test, with post hoc comparisons using Dunn’s method. Objective data were tested for normality using a Shapiro–Wilk test and analysed using restricted maximum likelihood (REML) mixed effects models with time as a random factor, foal as a fixed factor and post hoc testing by Tukey’s method. Non-normal data were log-transformed prior to analysis. All analyses were performed using GraphPad Prism version 10.0 for Windows (GraphPad Software, Boston, MA, USA, www.graphpad.com). 

## 3. Results

### 3.1. Behavioural Observations

The novel behavioural ethogram developed for the current study performed well, with minor amendments after F1 and F2 to better document locomotion, head jerking or abnormal positioning of the head, and issues relating to circuit leakage which were initially recorded as ‘other’ observations. All foals tolerated sedation and all interventions well. There was no evidence of epistaxis during or immediately subsequent to use of the nasal prongs, and no nasal discharge was evident subsequent to intervention. Foals appeared to be moving air equally and appropriately through both nostrils when examined on the day following interventions. All foals displayed some ‘distress’ behaviours during T1, the first 5 min after separation from their dam (Figure 4). There was some variability between foals with respect to the behaviours exhibited, including some behaviours that were expected to indicate discomfort with the prongs noted prior to placement of the PDI. Foal 3 was notably different from his peers, demonstrating somnolent behaviour on multiple occasions during T1, including one episode of recumbency. Two mares vocalized repeatedly during T1 and were sedated. Other mares stood quietly or could be heard eating in the background of the video recordings. 

Placement of the nasal prongs met with predicted avoidance behaviour, but this was ameliorated by cradling the head under the arm and by restraining the foal against the wall. The application of lignocaine gel prior to probe placement also appeared to be useful to decrease irritation during placement of the prongs. In general, once the prongs were in place, most foals appeared to rapidly become adjusted to their presence and appeared to tolerate the apparatus well. Four foals tolerated the prongs without sedation for the full five-minute intervention at T2. Two foals (F1 and F5) demonstrated persistent behaviours associated with discomfort (specifically head shaking of increased frequency and duration and attempts to remove the equipment, and were deemed intolerant of the prongs without sedation, although F1 subsequently tolerated the prongs after sedation. Behaviours consistent with minor distress or discomfort were demonstrated by all foals during T2, but the intensity and duration of these behaviours were mild. 

Somnolent behaviour was observed for F2 and F6 following sedation (T3), and this behaviour appeared potentiated in foals wearing nasal prongs in T4 (Figure 5; Appendix A). Sedation scores were low (maximum 9, of a possible score of 12) and, as expected, were significantly increased in foals following sedation at T3 and T4 (Figure 6). Although a significant time effect was observed (*p* = 0.045), post hoc comparisons were not significant. One foal (F5) remained refractory to prong placement after sedation but was left alone for 3 to 4 min at T4 and was subsequently tolerant of prong placement for spirometry and for NIV delivery. All foals tolerated NIV at PS and PEEP values of 2, 4, 6 and 8 cmH_2_O. Foal 3 demonstrated incremental agitation during NIV and was removed from the study after T8; F4 also demonstrated agitation during the final minute of T8 and was also removed from the study at this time. The first foal evaluated (F1) demonstrated a change in the pattern of respiration (erratic use of abdominal muscles) during T8. As respiratory distress could not be excluded as contributing to this behaviour, NIV was discontinued for this foal at this time. There was no increase in respiratory rate or other evidence of dyspnoea during NIV or subsequent to removal of bi-nasal prongs in this foal, and no other foal evidenced respiratory distress at any time. All foals were reunited with their dam following removal of all equipment and recovered rapidly from sedation. 

### 3.2. Physiological Changes

Heart rates during T1 ranged from 85 to 125 bpm, and significant differences were not observed during any intervention or observation period (Appendix A). Respiratory rate (RRobs) decreased progressively throughout the study (Appendix A), consistent with observed changes during spirometry and ventilation, discussed below. Mean RRobs in sedated foals was significantly less than that observed in unsedated foals at both T3 and T4. Subjectively, the depth of respiration appeared to increase throughout the study period. 

### 3.3. Spirometry Changes

Mean (95% CI) tidal volume determined using the mask PDI in standing, unsedated foals at T1 (618 mL, 488–747 mL) was not significantly different to values obtained from sedated foals at T3 (628 mL, 533–722 mL). As was observed for RRobs, Rf was lower through the nasal prongs than through the mask, and differences were significant for sedated foals (Figure 7). Nasal prongs were associated with significantly reduced Vt, and minute ventilation in sedated foals at T4 (15 L, 11–19 L; mean, 95% CI) was significantly lower than observed at T1 (33 L, 25–41 L; *p* < 0.001), T2 (25 L, 18–31 L; *p* = 0.024) or T3 (28 L, 23–34 L; *p* = 0.001). Inspiratory and expiratory flows were lower through nasal prongs; minimum O_2_ concentrations were lower; and maximum CO_2_ concentrations (EtCO_2_) were higher, associated with spirometry performed using nasal prongs (Figure 7). 

### 3.4. Non-Invasive Ventilation

NIV was discontinued after T8 (PS 8 cmH_2_O, PEEP 8 cmH_2_O) for F1, F3 and F4 based on behavioural responses; the remaining foals completed all five NIV interventions. Progressive increments in PS and PEEP were associated with decreased RRvent, but the effects were not significant when corrected for unequal variance (Figure 8). Minute ventilation (VEmin) decreased at T8 (8/8 cmH_2_O) although, again, differences were not significant. Inspiratory and expiratory volumes (Vti and Vte) and the ratio of these values (Vte/Vti) were log-transformed for analysis. A significant treatment effect was observed for Vti (*p* = 0.031), but post hoc comparisons were not significant (Figure 8). Although differences in Vte and Vte/Vti during NIV were not significant after correcting log-transformed data for unequal variance, both Vte and Vte/Vti progressively reduced with increasing pressures. Marked disparity between Vte and Vti (>1 L) was apparent at T6 (PS/PEEP 4/4 cmH_2_O) for F3 and F4. At this time, the ventilator readings indicated negligible expiratory return (low Vte values) and the Vti values were much greater than the corresponding tidal volume results at T1. Similar changes were observed for Vti for F5 at T8 (8/8 cmH_2_O). Inspiratory and expiratory times (Ti and Te) did not vary significantly during ventilation at lower pressures, but markedly increased Ti (typically >1.2 s) was observed from T6 For F1, F3 and F4, and a long expiratory phase was observed for F5. An audible expiratory leak was appreciated for F1, F2 and F4 during early NIV interventions and, at T8 for F2 and F5, with obvious leaking around nasal prongs during expiration. The one foal that tolerated all ventilator settings well (F6) had Vte/Vti of approximately 1 (i.e., was able to maintain Vte), Te substantively greater than other foals and Ti < 1 s, during each intervention (Figure 8). The ventilator settings resulted in appropriate and significant effects on Pmean and Ppeak (both *p* < 0.001, Figure 8). The intrinsic PEEP (autoPEEP) increased with the PS and PEEP settings to 6 cmH_2_O and then appeared to plateau (Figure 8).

## 4. Discussion

Contrary to our expectations, foals were remarkably tolerant of the prototype bi-nasal prong PDI evaluated in the current study. Four of six foals tolerated the prongs for the full five-minute intervention period without sedation and with minimal adjustment of equipment. Five of six foals tolerated the device following light sedation. Foals demonstrated mild and comparatively infrequent attempts to remove the prongs. Differences in RRobs were not evident, associated with the presence of bi-nasal prongs, although there was a tendency for slower, deeper breaths when the prongs were in place at T2 and T4, relative to comparative observations in foals without prongs (T1 and T3). 

This is the first time that NIV has been evaluated in standing, lightly sedated foals. All foals tolerated NIV delivery to PS and PEEP values of 8 cmH_2_O following light sedation, although behaviours consistent with discomfort increased as pressures increased. Consistent with previous studies in foals with pharmacologically induced respiratory insufficiency, NIV in the current study was associated with a dose–response increase in Vti and decreased RR, such that inspiratory minute volume did not change. In previous studies in recumbent foals, this change in respiratory function has been associated with decreased WOB and increased efficiency of gas exchange [6,7] due to increased lung volume, improved pulmonary aeration and enhanced V/Q matching [8]. Measures of gas exchange or pulmonary efficiency were not assessed in the current study of healthy, minimally sedated foals, where the primary research goals were related to assessments of tolerance to bi-nasal prongs and NIV. 

In the current study, Vti increased as the PS and PEEP settings increased, with observed values greatly exceeding Vt observed prior to NIV and greater than comparative results in previous studies using similar PS and PEEP settings in intubated foals [8]. In contrast to our previous studies, there was no concomitant increase in Vte, with values in the current study decreasing as pressure settings were increased. Whereas in intubated foals in our previous study [8], the observed Vte/Vti remained stable at a value approximating 1 (mean Vte/Vti 1.0 at 4/4 cmH_2_O, 95% CI 0.98–1.0), in the current study, at these pressure settings, the ratio was much less (mean 0.39, range 0–0.80), and a marked and incremental disparity was evident between Vte and Vti in five of the six foals as NIV pressures were increased. These findings are consistent with air leaking around the nasal prongs, as is inevitable during NIV [16]. During NIV in the current study, the exhalation valve is closed until airway pressure becomes 1 cmH_2_O higher than the desired PEEP. At higher pressure settings, therefore, increased inspiratory time (and hence Vti) are necessary to reach this opening pressure and allow expiration. This is exacerbated due to increased leak, with adverse effects on patient comfort and tolerance of NIV. Optimization of PDI fit is essential to the success of NIV. Leaks have been deliberately incorporated into NIV circuits to ameliorate CO_2_ accumulation within equipment dead space [17], but excessive leak contributes to patient intolerance and ventilatory failure due to patient-ventilator asynchrony [18,19]. 

Expiratory limitations have been recognized in foals associated with hypercapnia during NIV [6,7], and this has been interpreted as being due to increased intrinsic PEEP (PEEPi) and dynamic hyperinflation. Also termed autoPEEP, PEEPi is defined as the difference between measured PEEP and configured PEEP. Increased PEEPi predisposes to increased WOB, barotrauma, haemodynamic instability and difficulty triggering the ventilator [20]. In the current study, PEEPi increased and plateaued at higher NIV pressures, despite the observed leak associated with bi-nasal prongs and no concomitant increase in Te; that is, the leak was insufficient to prevent increasing PEEPi at higher pressures. Increased PEEPi was not observed in intubated foals in a previous study ventilated using similar PS and PEEP settings, a finding attributed to increased Te [5]. 

At increased pressures and associated increases in Vti and Ti, most foals in the current study demonstrated behaviours that necessitated discontinuation of NIV. Most commonly these changes were head shaking, abnormal head positions and/or attempts to remove the PDI. However, compromised neonates may be incapable of demonstrating such behaviours, necessitating objective measures to optimise ventilator settings. The presence of leaks decreases the sensitivity and responsiveness of ventilator effort-sensing capabilities, resulting in triggering delays or missed triggers, flow delivery mismatches and increased patient discomfort [21]. Missed triggers were evident on review of ventilator graphics in the current study, and marked differences in Vte/Vti and/or increased Ti preceded behavioural changes necessitating discontinuation of NIV. As noted above, failure to trigger the expiratory phase of the respiratory cycle was likely associated with increased leakage and prolonged Ti and consequently with the supraphysiologic Vti values recorded by the ventilator in the current study. Only the final foal, F6, demonstrated a response to ventilation approximating that seen in intubated patients, and it is interesting to speculate whether this might have reflected individual differences in tolerance or improved fitting of the PDI in this individual. 

Contrary to our hypothesis, the current study has demonstrated differences in spirometry between mask and nasal prong PDIs or, more likely, to the equipment necessary to adapt the nasal prongs to our respiratory flow head. Mask spirometry demonstrated that standing, unsedated foals in the current study had Vt of approximately 12 mL/kg, consistent with previous studies [8]. Spirometry using the bi-nasal prongs demonstrated adverse effects on respiratory function, including significantly reduced Rf evident at T4 (sedated with prongs) than evident when wearing prongs but not sedated at T2, or when sedated at T3 and undergoing mask spirometry. As these differences were not apparent for RRobs, it is likely the observed changes were associated with the increased dead space and resistance in the adaptor used to perform spirometry in foals with nasal prongs, rather than being due to the nasal prongs themselves. Differences were also apparent between tidal volume, inspiratory and expiratory flows, and gas composition (minimum O_2_ and maximum CO_2_ concentrations) from foals wearing nasal prongs at T2 and T4 than was evident for foals during mask spirometry. The changes in gas composition, in particular, are likely to represent an accumulation of expired air within the equipment dead space of this apparatus, particularly with the use of the adaptor to permit spirometry. Ideally, decisions to implement respiratory support and the interpretation of patient response to treatment, should be guided by objective measures of respiratory function [22,23], and spirometry is an attractive option for the characterization of respiratory mechanics in foals. Our data suggest that spirometry should be performed using a mask PDI or otherwise incorporated into the ventilator circuit for accurate measurement of lung mechanics. In addition to the assessment of ventilator data, current recommendations for monitoring of NIV in human patients include clinical assessment, arterial blood gases, transcutaneous pulsed oxygen saturation, transcutaneous capnography [24,25] and assessment of pulse oximetry/FiO2 ratio [26]. In equine [27,28] and human patients [29], the use of electrical impedance tomography is recommended to continuously visualize lung function and to instantly assess effects of therapeutic manoeuvres on regional ventilation distribution.

Somnolent behaviour, defined in the current study as a progressive decrease in alertness with concomitant narrowing of the palpebral fissure and lowering of the head, was exhibited by one foal (F3) prior to and following sedation (T1 and T2), and by five of six foals during sedation whilst wearing the bi-nasal prongs PDI or during NIV. Similar observations have been made by the authors of clinical cases during NIV (unpublished) and were considered in compromised neonates to possibly be a behavioural response to decreased work of breathing. Whilst a physiological response to altered respiratory mechanics is plausible (although not characterised), in the current study, it is possible that the observed changes reflect the effects of sedation and/or progressive comfort with an unfamiliar environment. Alternatively, the presence of a bandage around the chest (covering the ECG electrodes) might have contributed to the observed somnolence, as previously reported [30].

Although useful to assess tolerance of NIV methodology, behavioural changes are subjective and not without limitations. Behavioural changes in the current study were documented in real time using a novel ethogram to standardize and reduce the subjectivity of observations and were then validated by independent review and retrospective evaluation of the video recordings. The ethogram used in the current study has not been validated and was based only on the frequency of observed behaviours. The intensity of behaviours was not addressed, and it is possible that some behaviours might be more sensitive indicators of distress and discomfort than others. Although the retrospective assessments were performed in random order, interventions such as bi-nasal prongs and NIV were evident in the video, and the foals were known to the researchers, precluding a blinded assessment. The sequential study design did not allow us to control for factors such as accommodation to a novel environment or for time effects on sedation.

The use of a small number of healthy, lightly sedated foals is likely to have limited our ability to assess the effect of NIV on lung function, and objective or invasive measures of lung function were largely outside the scope of the current study. The amount of leak in the current system was difficult to quantify, although comparisons with a previous study of intubated foals using the same equipment and similar pressure settings were useful. Further studies are required to quantitate the leak associated with nasal prongs in foals and to optimize prong placement to minimize risks of leak or damage to nasal structures. The use of sectional imaging to assess prong fit might be useful in this regard. Although the duration of respiratory support offered to compromised foals (days to weeks) might be less than that attempted for human neonates (weeks to months) and no foal in the current study demonstrated evidence of nasal trauma, this is a frequent complication associated with the use of nasal prongs in human neonates [31,32]. Prolonged placement of nasal prongs in compromised foals is likely to be associated with similar risks of nasal trauma.

## 5. Conclusions

The current study demonstrated that foals were remarkably tolerant of nasal prongs and that healthy, standing, lightly sedated foals were amenable to both short-term placement of bi-nasal prongs and to the delivery of NIV. Higher pressure settings were associated with behavioural changes and ventilator data, including increased disparity between Vti and Vte associated with longer Ti and expiratory leak. These changes were predictive of ventilation failure and demonstrate the importance of characterizing the amount of leak present during NIV. Although increased Ti and Vti might be indicated for management of a restrictive respiratory profile [33], the findings from the current study suggest that excessive leak might be anticipated when Ti >1.2 s in foals, when there is disparity of >1 L between Vti and Vte, or when supraphysiologic Vti is delivered. Behaviour changes observed in healthy foals in the current study were relatively subtle and might not be demonstrated by hospitalized foals with respiratory and/or neurologic dysfunction. Assessments of respiratory mechanics by spirometry in spontaneously breathing foals should use a mask-to-spirometer interface, and further studies are required to ascertain whether the observed leaks associated with the bi-nasal prong PDI evaluated in the current study compromise the possible benefits observed during NIV in foals with respiratory insufficiency.

The suitability of bi-nasal prongs for the delivery of NIV and potential adverse effects of leakage should be further evaluated by studies assessing the effect of this PDI on gas exchange as a more sensitive indicator of pulmonary function. Ventilator values and graphics are likely to be useful in monitoring the response of sick foals to NIV but require further characterization in equine patients.

## Figures and Tables

**Figure 1 animals-14-00865-f001:**
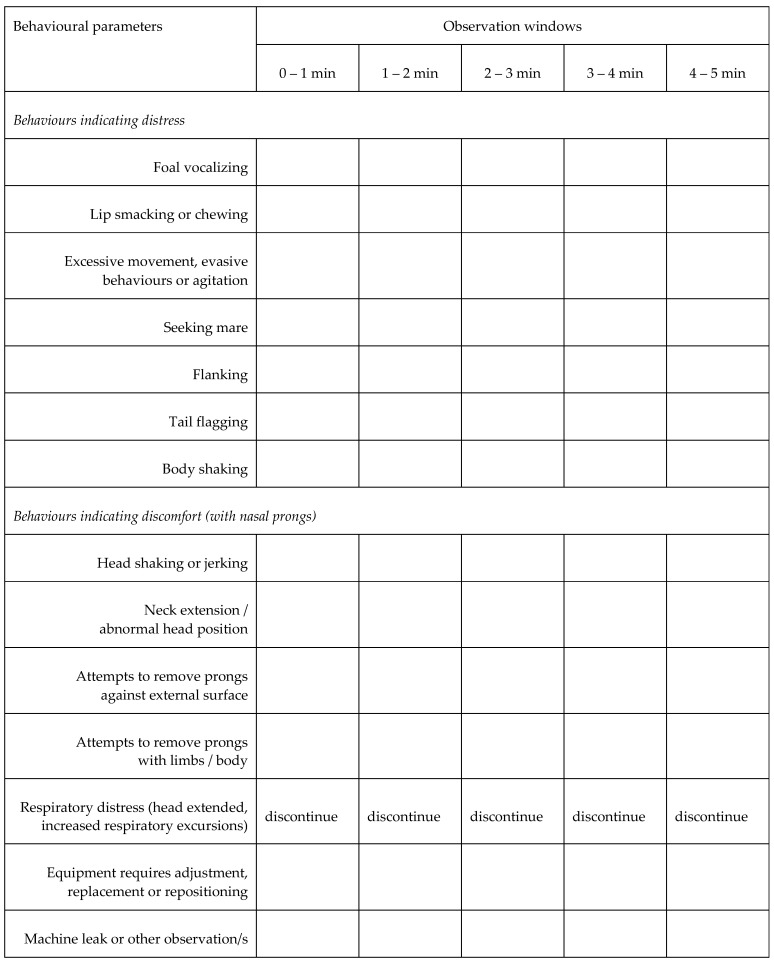
Behavioural ethogram for assessment of response to sequential interventions. The frequency of each identified behaviour was recorded during consecutive 1 min observation windows during each intervention for each foal by direct observation and subsequently by independent review of video recordings.

**Figure 2 animals-14-00865-f002:**
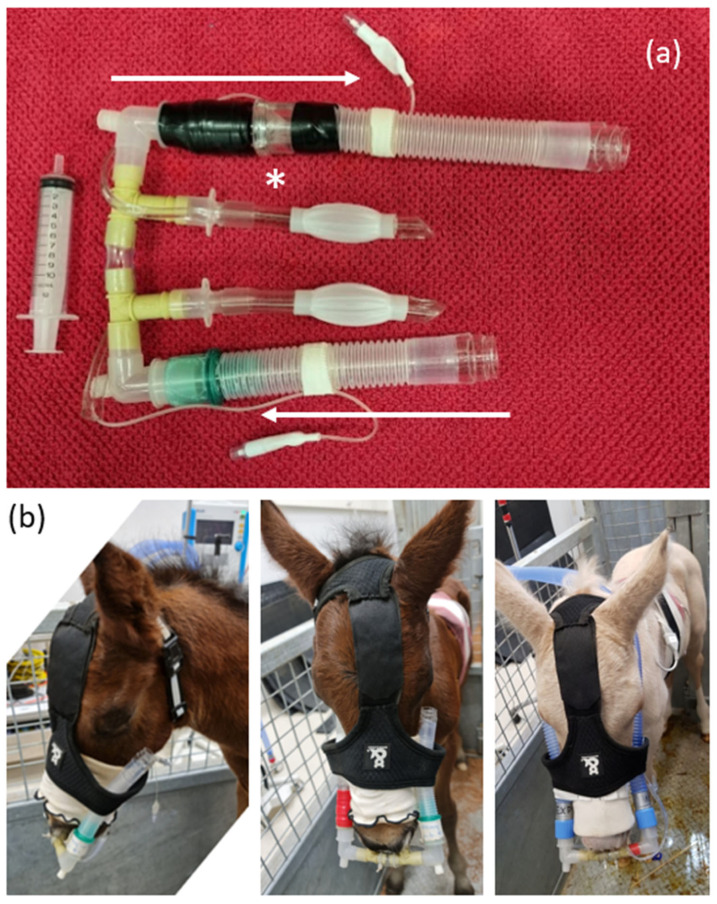
Prototype bi-nasal prongs evaluated in the current study. (**a**) Prongs with ‘dummy’ tubing for T2 and T4. White arrows indicate air flow, achieved by inclusion of a one-way valve (*). (**b**) Nasal prongs worn by F6 during T2 showing dummy tubing used to secure prongs in place using a modified harness (**left** and **middle** images). Non-invasive ventilation was delivered by connecting ventilator tubing directly to each elbow adaptor (**right** image), following removal of the dummy tubing and expiratory valve. The expiratory limb was placed to the right of the foal’s face.

**Figure 3 animals-14-00865-f003:**
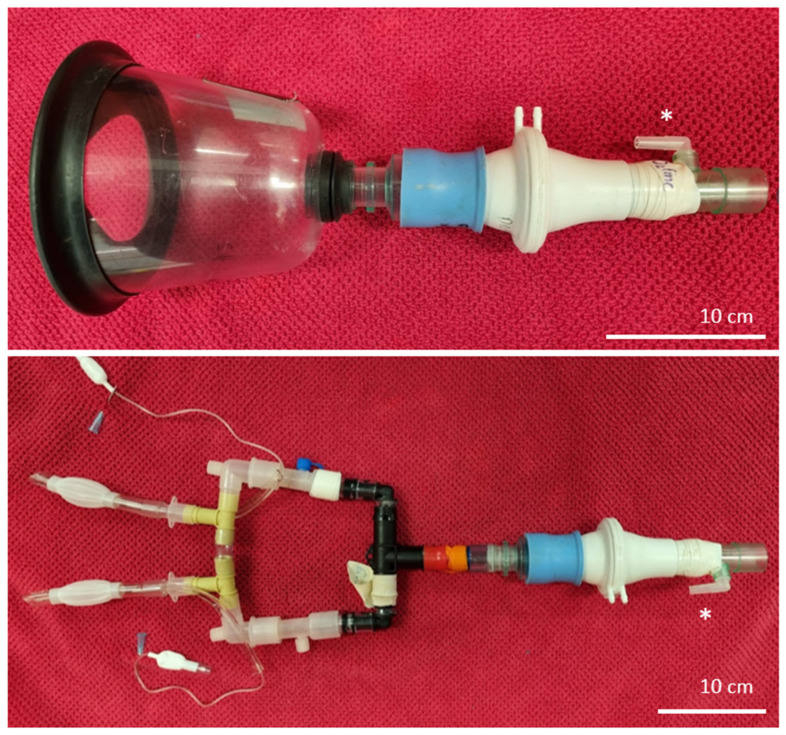
Spirometry apparatus for foals without nasal prongs (**top**) showing spirometry flow head and gas sampling port (*) connected to a large animal anaesthesia mask. Lung function associated with bi-nasal prongs was assessed by connecting the elbow adaptors to a short T-piece adaptor as shown (**bottom**).

**Figure 4 animals-14-00865-f004:**
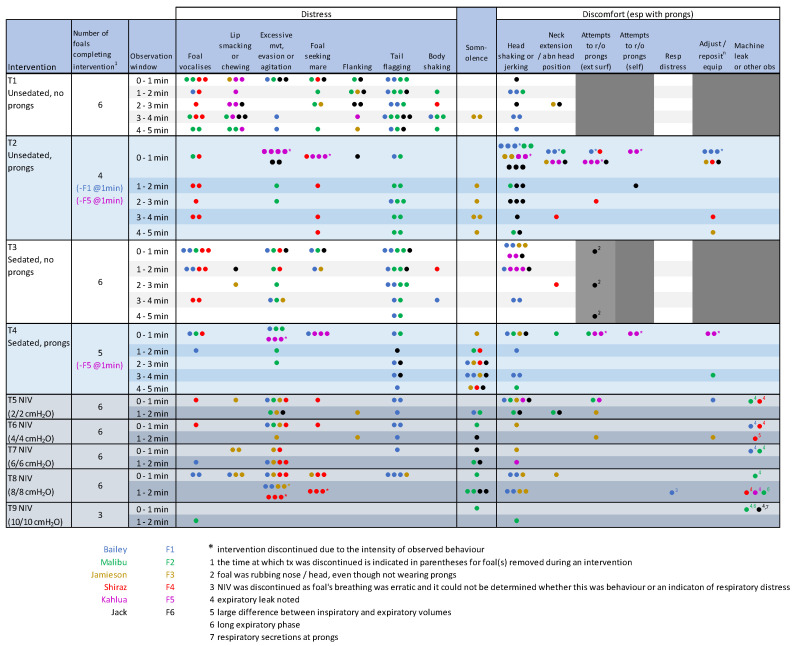
Composite ethogram showing incidence of behaviours indicative of distress or discomfort relating to bi-nasal prongs or other interventions. Within each observation window, the number of times a behaviour was observed was recorded on the ethogram for each foal, with combined data presented below. Individual foals are indicated by colour (F1 blue, F2 green, F3 yellow, F4 red, F5 purple and F6 black), with the number of dots indicative of the number of times a behaviour was observed.

**Figure 5 animals-14-00865-f005:**
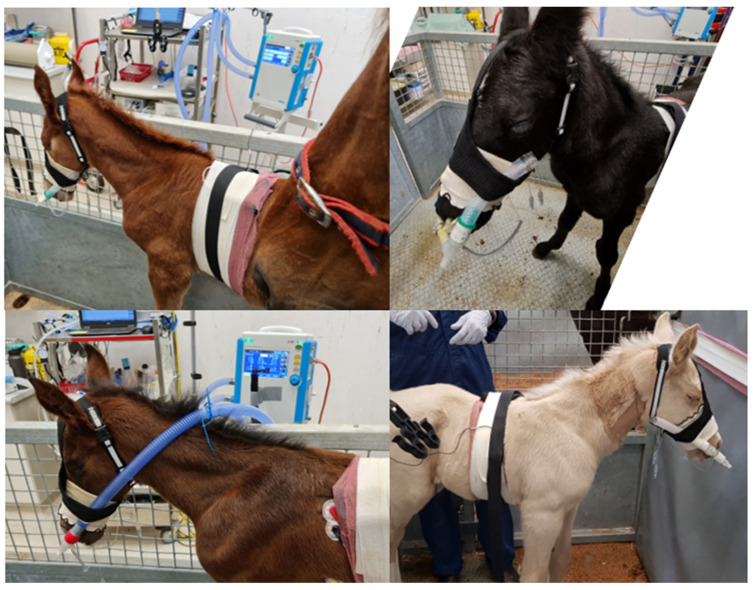
Clockwise from top left, F3, F4, F1 and F6 demonstrating somnolent behaviour during T4 (sedated, bi-nasal prongs with dummy tubing) or during NIV (F6, bottom left).

**Figure 6 animals-14-00865-f006:**
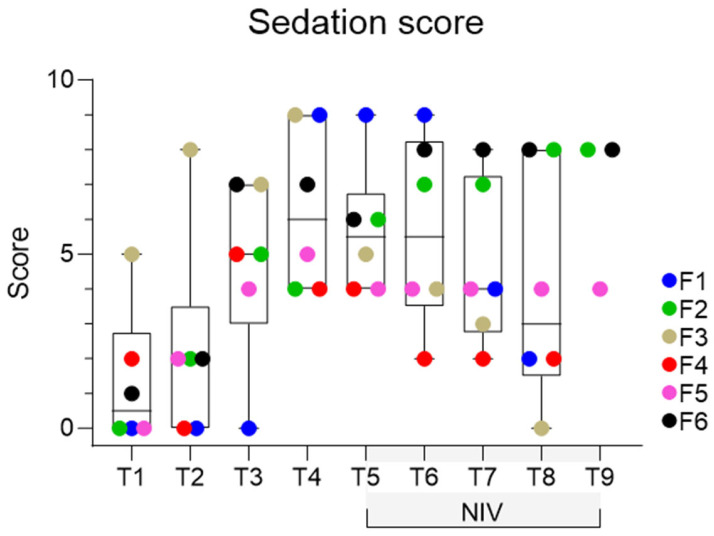
Sedation scores during sequential interventions, as described in Table 1. A significant time (treatment) effect is present (*p* = 0.045), although post hoc comparisons were not significant. Results are shown as median (horizontal line), inter-quartile range (box) and range (whiskers), with all data points shown and individual foals identified by colour, as shown. As only three foals completed T9, individual data points for these foals are presented, and this time point was excluded from analysis.

**Figure 7 animals-14-00865-f007:**
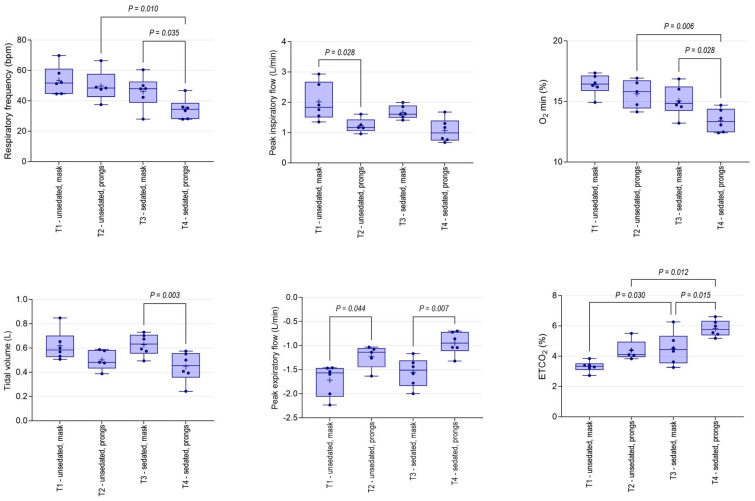
Spirometry and gas analysis variables in unsedated and sedated foals associated with mask or bi-nasal prongs as patient-device interface. Results are shown as mean (+), median (horizontal line), quartile (box) and range (whiskers), with all data points shown. Significant differences between treatments are shown.

**Figure 8 animals-14-00865-f008:**
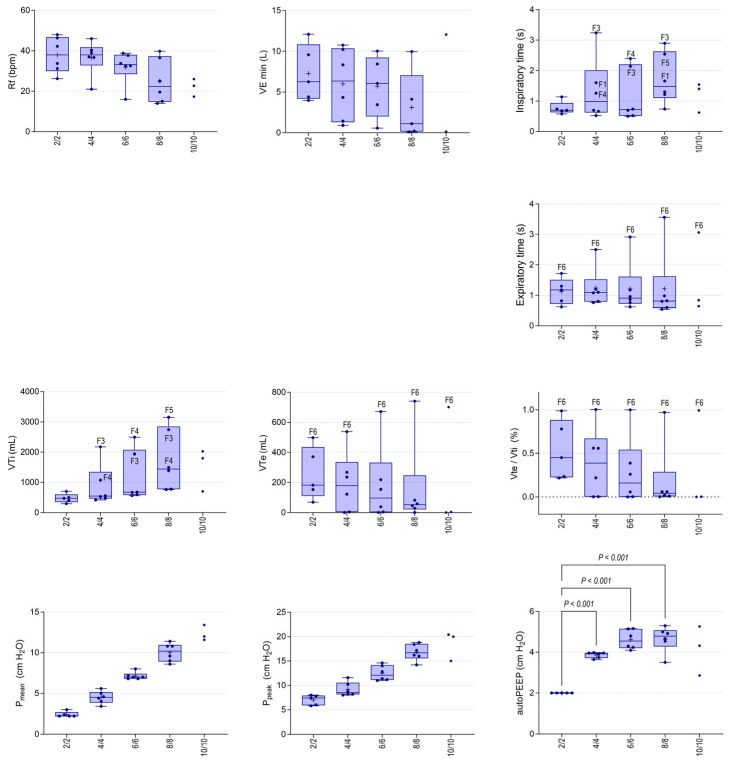
Ventilator variables during NIV delivered by bi-nasal prongs in standing sedated foals associated with different pressure support and positive end-expiratory settings (PS/PEEP). Results are shown as median (horizontal line), quartile (box) and range (whiskers), with all data points shown and mean (+) indicated for normally distributed data. Outliers are identified (F3, F4, F5 and F6) as appropriate, and significant differences between consecutive NIV pressure settings are shown for mean and peak airway pressure, and significant differences in intrinsic (auto) PEEP, compared to 2/2 cmH_2_O, are shown. Rf, respiratory frequency; VEmin, minute expiratory volume; VTi, inspiratory tidal volume; Vte, expiratory tidal volume; Pmean, mean airway pressure; Ppeak, peak airway pressure; autoPEEP, auto (intrinsic) peak end-expiratory pressure; abscissa values are given as pressure support and PEEP settings corresponding to T5 to T9).

**Table 1 animals-14-00865-t001:** Order and duration of sequential interventions for each foal. Spirometry data were collected for 30 to 60 s after completion of observations during T1 to T4, as shown. Foals were allowed to recover for 1 to 2 min subsequent to equipment changes or sedation, prior to collection of subsequent behavioural data.

Time	Intervention	Duration (min)	Approximate Elapsed Time (min)
T1	Unsedated, no PDI	5	5
	Mask spirometry		6
	Place nasal prongs, allow to settle		8
T2	Unsedated, nasal prongs in place	5	13
	Nasal prong spirometry		14
	Remove nasal prongs		
	Sedate, allow to settle		16
T3	Sedated, no PDI	5	21
	Mask spirometry		22
	Place nasal prongs, allow to settle		24
T4	Sedated, nasal prongs in place	5	29
	Nasal prong spirometry		30
	Connect NIV circuit, allow to settle		32
T5	NIV 2/2 (PS ^1^ 2 cmH_2_O, PEEP ^2^ 2 cmH_2_O)	2	34
T6	NIV 4/4 (PS 4 cmH_2_O, PEEP 4 cmH_2_O)	2	36
T7	NIV 6/6 (PS 6 cmH_2_O, PEEP 6 cmH_2_O)	2	38
T8	NIV 8/8 (PS 8 cmH_2_O, PEEP 8 cmH_2_O)	2	40
T9	NIV 10/10 (PS 10 cmH_2_O, PEEP 10 cmH_2_O)	2	42

^1^ pressure support; ^2^ positive end-expiratory pressure.

**Table 2 animals-14-00865-t002:** Criteria used to determine an overall sedation score for each foal during each intervention window.

Score	Head Height above Ground	Response to Tactile Stimuli	Response to Auditory Stimuli	Postural Stability
0	Head carried in a neutral, alert position	Intense movement of body/body part, steps away	Appropriate, rapid movement of head, neck and body, evasive action	No swaying, weight bearing on all limbs or appropriately resting one limb
1	Neck inclined above horizontal, but head not fully elevated	Moderate movement of body/body part, without evasive action	Rapid or slightly reduced movement of head, body or ears	Nil or minimal swaying, nil or minimal postural adaptation
2	Head lowered, neck horizontal or below	Slight movement of body/body part, minimal response to touch	Slow and limited movement of head, neck or ears	Moderate swaying, base-wide stance and/or limb/s malpositioned
3	Head lowered to <25% of full height	No response	No response	All above plus intense swaying/instability, risk of falling

## Data Availability

The data presented in this study are available on request from the corresponding author.

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
