# Peer review of "The Use of Bi-Nasal Prongs for Delivery of Non-Invasive Ventilation to Foals"

_animals, 2024, doi:10.3390/ani14060865_

Round 1
Reviewer 1 Report
Comments and Suggestions for Authors
This is a very interesting study which provides further information to enable improvement of respiratory support in foals.
Specific comments:
Title: I wonder if the title would be better stated as simply – ‘the use of’ binasal… or similar, as not all foals tolerated the intervention, and it underplays the challenges that were identified using the prongs in their current format.
Abstract is clear and concise
Line 101: do you have any other information to help validate your ethogram?
Line 143: were cuff inflation pressures measured? Or volume of air used in cuff recorded?
Figure 1: the positioning of the inspiratory and expiratory limbs via the elbow adaptors meant that they were separated by a reasonable distance- do the authors think that this had an effect on gas mixing/ provision of consistent flow/ FiO2?
Line 250- : Could the authors provide video footage of the NIV in action – both with good tolerance and poor tolerance in addition to the somnolence clips in the supplementary information?
Figure 5: Right hand side of figures is not visible
Line 359: Did VEmin not decrease at 8 and 10cm timepoints?
Line 367: Why was the effect greater in NIV than in intubated foals?
Line 374: Could the authors use 3D imaging with the prongs in place to help improve design to reduce leakage?
Line 388: Could the authors comment on whether the tolerance was mainly related to increasing disparity of the ventilator functions rather than the actual increasing airway pressure?
Line 414: Do these problems invalidate the spirometry results?
Line 454: Could the authors add a comment about the small number of foals included and effect on results?
Were any efforts made to assess for nasal trauma in foals after use of the prongs? Any other adverse effects such as nasal breathing, swelling etc. Would endoscopy be useful to assess this. Do the authors think that prolonged use of the prongs would be problematic?
Author Response
Our thanks to reviewer 1 for their comments. We have addressed each comment in the attached document.

Reviewer 2 Report
Comments and Suggestions for Authors
Dear Authors,
The original paper “« Tolerance of binasal prongs for delivery of non-invasive ventilation to foals » could add some useful information about the feasibility and tolerance of non invasive ventilation in neonatal foals using a new patient-device interface. The paper is well written, nevertheless, I have a few remarks and suggestions I would like the authors to consider.
Abstract
L19: “in standing lightly sedated foals” please specify “in 6 healthy standing …foals”
1.Introduction
L70:”the use of nasopharyngeal prongs has been described”: In the article of Floyd et al. EVJ 2022, I think they use thoracic chest drains as nasopharyngeal “prongs” but the device was not designed to create a seal for high flow oxygen therapy and let 50% of the nasal diameter free. Please note this difference with your PDI.
2.Materials and methods
2.2 Experimental design, physiological and behavioural responses
L98 This part is difficult to follow before to read the table 1, maybe a short description of the sequential interventions could introduce the experimental design at the beginning of this part in the text before to describe the details of behavioural evaluation etc.
L101-107 and L114 Table 2: “Foal behavioural responses… a sedation score” Please can you add a reference for this ethogram in foals (I could not find something similar in foals in the literature, but maybe it was a modified Equine discomfort ethogram designed for adult horses?)
L104-106: “video footage… and assign a sedation score”: The sentence is confusing for the sedation score, please specify if the sedation score is attributed later and not in real time including the response to tactile stimuli and to auditory stimuli?
L111 table 1: maybe the total duration of the experiment and the time when spirometry was used could be inserted into the table to increase readability.
L132-134: “Interventions were discontinued if the foal… respiratory distress” please specify how did you use the ethogram in this way to decide if the treatment should be stopped for a foal or what level of discomfort is acceptable (score?)
2.3 Instrumentation
L171-173: “Non invasive ventilation…(Figure 1).”Could you specify the gas inhaled (room air?) the ventilator modalities and setting (pressure/volume controlled ventilation? Patient triggering for starting and ending of inspiration) for people who are not very familiar with this treatment in foals
2.4 Spirometry and gas analysis
The sequencing of events is not clear, could you start this part by a sentence summarizing the events: spirometry was performed at T1 (mask without sedation), T2 (nasal prongs without sedation), T3 (mask after sedation) and T4 (nasal prongs after sedation) after 5 minutes observation (T1,T3) or after completion of 5 minutes intervention windows (T2,T4).
3.Results
3.1 Behavioural observations
Table 4: ‘*Behaviours that justified discontinuing intervention” Could you add in the text or in the table what criteria you have used (expression of more than 2 times the same distress/discomfort behaviour during one minute? Or the sum of these behaviours?) because it seems different along with the sequential interventions
L262-270 : «Foal 3… from sedation.” Although time effect was not statistically significant in post-hoc comparisons in you study, a decrease in sedation score with time is expected with wide individual variations as shown by figure 4. Behavioural observations at T8 and later could also be related to the sedation score especially for foals 1, 3 and 4 for which the intervention had to be discontinued after T8 and which showed a low sedation score at this moment.
3.3 Spirometry changes
L304 “mask sedation” should be change for “mask interface” for example
L315 figure 5: the graphs are not legible for O2min and ETCO2
3.4 Non invasive ventilation
Figure6: please note the meaning of the abbreviations used on the graphs
4.Discussion
To my mind, the discussion appear a bit confused and could be more focused on the 3 hypothesis and aims of the study:
1/tolerance to nasal prongs: assessed by behavioural observations(subjective) and physiological parameters (HR, RRobs and respiratory effort (subjective)) with and without sedation: well tolerated (4/6 foals without sedation and 5/6 foals with sedation)
2/tolerance to NIV: assessed by behavioural observations (subjective), physiological parameters (HR, RRobs and respiratory effort (subjective)) and ventilator parameters only with sedation (before to start NIV which lasts 30minutes or more?) without any blood gas analysis or pulmonary function assessment : well tolerated until T8, but air leak during expiration for 5/6foals including 1 case of respiratory distress (dyssynchrony?)
3/Spirometry differences between mask and nasal prongs with and without sedation: not conclusive because of increased dead space with nasal prongs-spirometer interface compared to mask-spirometer interface
L349 “full 5 minutes intervention period with without sedation” suppress the word “with”
L351-352 “All foals tolerated NIV delivery to PS and PEEP values of 8cmH2O.” specify “following light sedation”. Moreover the main drawback of this new device for foals could be discuss ie the occurrence of leaks during expiratory phase, potentially leading to trigger missing from the ventilator, dyssynchrony and discomfort, and how this prototype could be improved to better fit the nasal cavities of the foals keeping in mind the difficult balance between a tight PDI (more than moderate inflation of the cuffs >70%) and the risk of local complications (nasal irritation or even necrosis we can imagine) which could be explored in further studies
L359: During NIV, the effect of sedation with diazepam (0.1mg/kg) on ventilation parameters of these foals could be discussed as a confounding factor since this treatment could decrease the respiratory rate and increase the tidal volume to maintain the minute ventilation (after 0.2mg/kg of diazepam according to Sacks and Raidal frontiers vet sci 2023) as observed in your study
L429-440 “”Somnolent behaviour”, as previously reported” this paragraph could be shortened because in the absence of a control group (foals in the same conditions and duration of separation with their dam in a new environment without NIV), it is indeed difficult to conclude if this behaviour is really favored by the interventions (sedation, NIV or bandage)
L455: Concerning the limitations, the very short use of NIV in this study could be mentioned because it precludes the full assessment of tolerance and the occurrence of other complications well described with NIV in human (interface related complications including CO2 rebreathing, discomfort (device, pressure high or low), facial skin lesions (increase with NIV duration) nasal irritation, patient-ventilator dyssynchrony, air leak syndrome, airways dryness, and gastric insufflation (cf Carron et al. Complications of non invasive ventilation techniques in British J of Anaesthesia 2013 110 (6): 896-914) and discuss these potential complications in light of your and other studies (1 patient-ventilator dyssynchrony could be suspected for foal 1 at T8 in your study, 1 case of abdominal distension with high flow oxygen therapy in the paper of Floyd with a pressure of approximately 7cmH2O)
5.Conclusions
L459: “ nasal prong and to delivery of NIV”: “short term” should be added to the sentence because other complications typically appear after a few hours to a few days like nasal irritation and this point was not explored in this study
L466-468 “Behaviour changes …neurologic dysfunction” the monitoring should then include objective parameters (to assess pulmonary function/gas exchange) as it was already mentioned in the discussion
Author Response
Our thanks to reviewer 2 for their feedback. A response to each comment has been made in the attached document.
